# Prospects of Using Protein Engineering for Selective Drug Delivery into a Specific Compartment of Target Cells

**DOI:** 10.3390/pharmaceutics15030987

**Published:** 2023-03-19

**Authors:** Andrey A. Rosenkranz, Tatiana A. Slastnikova

**Affiliations:** 1Laboratory of Molecular Genetics of Intracellular Transport, Institute of Gene Biology of Russian Academy of Sciences, 34/5 Vavilov St., 119334 Moscow, Russia; tslast@genebiology.ru; 2Department of Biophysics, Faculty of Biology, Lomonosov Moscow State University, 1-12 Leninskie Gory St., 119234 Moscow, Russia

**Keywords:** targeted drug delivery, proteins, subcellular transport, cancer, protein engineering, multifunctional proteins

## Abstract

A large number of proteins are successfully used to treat various diseases. These include natural polypeptide hormones, their synthetic analogues, antibodies, antibody mimetics, enzymes, and other drugs based on them. Many of them are demanded in clinical settings and commercially successful, mainly for cancer treatment. The targets for most of the aforementioned drugs are located at the cell surface. Meanwhile, the vast majority of therapeutic targets, which are usually regulatory macromolecules, are located inside the cell. Traditional low molecular weight drugs freely penetrate all cells, causing side effects in non-target cells. In addition, it is often difficult to elaborate a small molecule that can specifically affect protein interactions. Modern technologies make it possible to obtain proteins capable of interacting with almost any target. However, proteins, like other macromolecules, cannot, as a rule, freely penetrate into the desired cellular compartment. Recent studies allow us to design multifunctional proteins that solve these problems. This review considers the scope of application of such artificial constructs for the targeted delivery of both protein-based and traditional low molecular weight drugs, the obstacles met on the way of their transport to the specified intracellular compartment of the target cells after their systemic bloodstream administration, and the means to overcome those difficulties.

## 1. Introduction

The effect of the vast majority of drugs is ultimately realized inside the cell due to their interaction with biomacromolecules. All drugs that have an effect on intracellular processes can be roughly divided into four main groups (Figure 1). Traditional low molecular weight medications, most of which are subject to the rules formulated more than 20 years ago by Lipinski et al. [1], can be attributed to the first group (Figure 1A). These substances are capable of nonspecific diffusion through the plasma membrane and exert their effect on the metabolic and regulatory processes within cells. Therefore, the distribution of such drugs is generally independent of the cell type. For various drugs and a number of disorders, the lack of cell specificity is not crucial; however, for the treatment of many diseases, it creates fundamental problems. These include, first of all, oncological diseases, in the treatment of which highly toxic antitumor drugs are used. 

Many drug delivery vehicles that facilitate penetration into cells, regardless of the cell type, are in fact close to this group. Such delivery systems include, for example, cell-penetrating peptides (CPP) [2], bacteria-based delivery systems [3], cell-penetrating poly(disulfide)s [4], mesoporous silica nanoparticles [5,6], micelles [7], and many others.

Low molecular weight substances recognized by special transmembrane transporters can be considered the second group of drugs for intracellular targets (Figure 1B). More than 400 such special transmembrane transporters are known up to now. These transmembrane transporters are necessary for cells to deliver substances that cannot freely diffuse through the membranes [8]. The variety of molecules that can be delivered into cells in this way is relatively small. This is due to the need to comply with certain requirements specified by the properties of the specific transporter. However, the development of drugs using such pathways into the cell is of great interest for the treatment of cancer, since intensively dividing malignant cells often exhibit increased expression of the transporters [9,10,11].

The third group of drugs exert their effect on the regulation of intracellular processes through cell surface receptors and pathways for further signal transduction (Figure 1C) [12]. Examples of such drugs are hormones and cytokines and their analogues, as well as mAbs [13] that bind to the surface cell surface receptors or ligands of those receptors [14]. 

Recently, more and more drug developments have begun to appear, allowing therapeutics to exert their effect inside the specific compartment of the specific cell type [15,16,17,18,19]. Such drugs can be attributed to the fourth group, which differ in their method of penetration into cells and their effects on intracellular processes from those mentioned above (Figure 1D). For example, they can combine both a means of delivery to the desired part of the target cell and a cargo drug. Many of these therapeutics are based on the synthesis of nanoparticles decorated with various functional components, including various ligands for internalized surface receptors (e.g., proteins, carbohydrates, and low molecular weight substances), CPPs, fragments of bacterial toxins for delivery to the cytosol, nuclear localization signals (NLS) for delivery to the nucleus, and polypeptide sequences for transport into cell organelles. Being rather large constructs, nanoparticles cannot freely penetrate into cells and so are forced to use either natural ways of transport of macromolecules in living systems or additional steps to disrupt membrane integrity. The latter itself is toxic to cells, leading to additional side effects. Frequently, peptides/polypeptides or their artificial analogues are used to either functionalize new kinds of nanoparticles [20] or to develop self-assembling nanoparticles [21]. Moreover, polypeptides themselves may possess the necessary medicinal and/or transport properties. Therefore, another direction to obtain effective and cell-type-specific constructs for drug delivery is the development of recombinant proteins. In addition, most intracellular regulatory pathways are regulated via protein–protein interactions, and a polypeptide seems the most appropriate candidate to interfere in this process [22]. Proteins in the right place can themselves serve as effective therapeutic agents. This can be achieved by delivering the necessary protein into the right cell. To date, considerable experience has already been accumulated in identifying amino acid sequences that are domains of proteins involved in intracellular regulatory and transport pathways. Therefore, an attractive feature of the protein-based approach is its ability not only to deliver convenient low-molecular drugs into the most appropriate cell compartment but also to incorporate an active principle directly into the developed construct. In the future, this may lead to the precise regulation of a pathologically altered process adapted for a particular patient. There are many obstacles to the implementation of such a direction, concerning both the existing barriers to the delivery of a suitable drug to the right place of the right cell and the lack of knowledge. Nevertheless, it is already possible to try to evaluate the necessary properties of such structures and the conditions that would allow them to be obtained. This direction is all the more important since 75–80% of protein targets are intracellular proteins that lack obvious sites for the action of small-sized drugs [23]. Alternatively, gene therapy approaches [24] as well as mRNA delivery [25] are also considered as means to provide the appearance of the right protein in the right cell. In this case, it is necessary to ensure the effective delivery of DNA or RNA to the nucleus or cytosol, respectively, of the target cell. The delivery of nucleic acids for therapeutic purposes is a separate area that is generally outside the scope of this review.

The current field of multifunctional protein-engineered therapeutics intended for selective drug delivery into a specific compartment of target cells, their prospects, and their limitations are summarized in this review. These constructs can carry either low molecular weight drugs or therapeutic amino acid sequences.

Cell and intracellular specificity, as the keys to selective targeting, will be discussed first. Then, we will shed light on the extracellular and intracellular barriers for therapeutics to overcome on their way to the intracellular target. This will be followed by a review of internalizable protein-based delivery systems and their intracellular fate after cell entry.

## 2. Specificity

### 2.1. Cell Specificity

To ensure the selectivity of the effect with respect to the cell type, the drug used must have the ability to recognize them. Optimally, the drug should identify the desired cell from its outside. This minimizes the side effects of drugs, preventing their action on non-target cells. Recognition can be achieved by binding to cell surface receptors. It is estimated that more than 1350 human genes encode surface cell receptors and provide regulation in the cell in response to external signals [26]. They make up about 40% of all membrane proteins with known functions [26]. Differences in the expressions and cell surface manifestations of these receptors determine the possibility of cell-specific action for biologically active substances that are not capable of entering cells by passive diffusion. These receptors are actively used to develop drugs that can affect the functioning of cells without penetrating them. This kind of drug is presented by their natural agonists as well as natural and synthetic antagonists, including blocking antibodies. Many of these substances enter cells through various types of receptor-mediated endocytosis after binding to their receptors. The internalization of active ligand–receptor complexes serves first of all to attenuate the signal entering the cell. This process can be also used for selective delivery of an active principle into cells, with several examples having already reached the clinical stage [27,28].

The greatest attention to the characteristics of the cells differentiating them from other cell types is paid in the development of anti-cancer drugs [29]. However, the treatment of malignancies is only one of many areas for which the cell-type-specific effects of drugs are significant. The rapid development of transcriptomics and proteomics methods has made it possible to identify differences between the proteomes of individual cell types in the body. The estimates of proteome specificity for 13,640 proteins of 29 tissues of the human body, which were performed recently by Wang et al., show the existence of at least 2000 “tissue-enriched” (defined as at least fivefold above any other tissue) proteins [30]. Although the proportion of such proteins is small for almost every tissue, their total number opens up a huge field for the specific impact on tissues. In addition, about 3000 proteins were classified by the authors as “group enriched” (defined as at least fivefold above any group of 2–7 tissues). Significantly, of the 486 drug target proteins found, only 34% are expressed in all tissues. At the same time, about 47% of those proteins belong to the “tissue-enriched” or “group-enriched” groups. These data emphasize the significance of the heterogeneity of drugs’ effects due to the uneven expression of targets in tissues. The largest proportion of tissue-specific proteins is observed among the G-protein coupled receptors (GPCRs), although the representation of such proteins is low. The uneven expression of surface cellular receptors across tissues opens up the possibility of tissue-specific drug delivery into the cell.

The actual differences in the content of proteins in cells are significantly greater, as each tissue usually consists of several basic cell types. There are many different cell types in the human body; however, their actual number remains a debatable issue [31]. Currently available cell databases give a different number of cell types from about 200 to a couple of thousand [31]. The understanding of the processes actually taking place in the body at the level of a single cell is rapidly being refined. An analysis of the available single-cell transcriptomics data conducted in 2021 for cells from 13 human organs, which were classified based on the gene expression profiles, showed the presence of 192 cell clusters [32]. Notably, less than 22% of the genes possess low specificity for cell type and only 11% of the genes were found to be expressed in each of the cell clusters. According to these data, about a quarter of all genes are specific to some cluster or group of clusters [32]. A recent paper announces “Tabula Sapiens”—a single-cell transcriptomic atlas containing data on nearly 500,000 cells from 24 different human tissues and organs [33]. The atlas shows the presence of at least 475 cell types in the human body. It should be noted that cells of the same origin, for example, macrophages or endothelial cells, differ in the profile of gene expression depending on the tissue. An additional contribution to the differences between cell types is made by alternative splicing, which may be cell-type-specific [33]. These data are of twofold importance in predicting the body’s response to drugs. On the one hand, this makes it possible to evaluate the distribution of drug targets across tissues and cell types. On the other hand, this enables us to create drugs or drug delivery systems that target precisely the cells for which the effect is required.

Moreover, approaches to the study of the spatio-temporal regulation of the proteome in the cell and its post-translational regulatory modifications are beginning to appear [34]. The principal possibilities of obtaining such information have recently been demonstrated by the example of changes in protein phosphorylation and changes in their intracellular localization after the addition of epidermal growth factor [34]. At the same time, more than 7000 proteins and 11,000 phosphorylation sites were monitored simultaneously in experiments on the HeLa cell culture. A comparison with a similar experiment conducted in vivo showed both the presence of a correlation with data on the cell culture and some differences that are anticipated when studying regulation systems at different levels of the living [34]. Such approaches, in addition to the already available detailed work on the study of many regulatory pathways, make it possible to assess how the cell as a whole reacts to the received stimulus.

It is noteworthy that information related to aging-dependent changes in gene expression by different cell types in the body is also beginning to emerge. The data were obtained for mice ranging in age from one month to old age [35,36]. Similar works on the human body are still more fragmentary; however, they are certainly important from the point of view of the peculiarities of treatment in aging [37]. 

Despite the vast amount of knowledge accumulated on the specificity of protein expression in the cell, in general, data on differences in the proteome in the body are currently far from complete. This is especially true for personal individual differences as well as differences at the level of a single cell. Moreover, as of 2021, the expression of 1421 predicted proteins coded in the human genome has not been detected yet, although this number is decreasing every year [38]. Nevertheless, progress in single-cell proteomics gives rise to hope that it will be possible to systematically predict the effect of an intracellularly delivered drug on a population of cells that express a particular receptor on the cell surface [39]. New and highly challenging techniques for spatial single-cell proteomics that are now extensively developed can broaden the horizons of precisely targeted cancer therapy opportunities [40].

It should be noted that, as in the case of specificity with respect to cell type, a significant part of the proteins has a predetermined intracellular localization, which can be illustrated by data from the Human Protein Atlas database (Table 1) [41]. This intracellular specificity is superimposed on cell specificity, which affects the process of drug delivery to intracellular targets. According to Human Protein Atlas estimates, the localization of about a quarter of intracellular proteins is confined to only one organelle, while only 7% of proteins are present in three or more different intracellular locations. It should also be taken into account that, during translation, almost all proteins appear either in the cytosol or in the rough endoplasmic reticulum (ER). It is noteworthy that the amount and localization of a significant part of the proteins is cell-cycle-dependent.

### 2.2. Subcellular Specificity

As a substantial proportion of drugs act or are at least more effective on specific intracellular targets, the recognition of the desired cell type should in many cases be followed by the subsequent intracellular transport of the delivered therapeutics to a target subcellular compartment [42]. Numerous therapeutics (which can either demand tissue/cell specific targeting or not) requiring organelle-level specificity are now being widely developed and extensively studied. These strategies are supposed to be promising modalities for the treatment of a broad spectrum of diseases, including first of all metabolic, neurodegenerative, and neuromuscular diseases, and cancer [43]. Almost any organelle present in cells can serve as a target for this or that therapeutics.

The nucleus represents one of the most obvious and widely studied subcellular targets for a number of drug classes. Gene therapy setting, where either genetic material or a gene editing system should be delivered to the nucleus, can be regarded as a canonical example of nuclear-targeted therapeutics. The other examples include but are not limited to photosensitizers [44,45], radionuclides [46,47], chemotherapeutics [48], and thermal ablating agents [49].

Being the cell’s “power station” and programmed cell death master regulator as well as a key player in many other vital processes such as cell differentiation and division, mitochondria represent an attractive target for various, especially anticancer, therapeutics. Mitochondria-targeted chemotherapeutics [50], nucleic acids [51], proapoptotic peptides [52], and agents for photodynamic [53], photothermal [54], and sonodynamic [55] therapy have been already successfully assessed on the in vitro and in vivo levels. The aforementioned approaches mainly aim at mitochondria-mediated apoptosis induction. In addition, a few completely different strategies, e.g., mitochondrial function restoration, for the treatment of neurodegenerative, cardiovascular, metabolic and other diseases linked with mitochondrial dysregulation are also under extensive development [56].

The restoration of proper functioning is also among the most widely utilized approaches in lysosomal targeting. This strategy is widely used for the treatment of lysosomal storage diseases, with a number of therapeutics already used in clinical settings for the treatment of several enzyme deficiency disorders (Gaucher disease, Fabry disease and some others) [57]. On the other hand, the lysosomal ability to degrade biologic molecules is utilized for the selective degradation of the target membrane proteins responsible for disease, particularly cancer pathogenesis and/or progression [58,59], and even to achieve efficient antigen presentation [60].

ER stress and the subsequent unfolded protein response supposed to underlie such pathologies as neurodegeneration, cancer, and diabetes represent an attractive target for therapeutic modulation [61]. To achieve this, various strategies for ER targeting have already been or are being developed [62].

Other cell organelles, e.g., Golgi apparatus, plasma membrane, and peroxisomes, are also used for precise targeting in the development of strategies for treating serious diseases, such as cancer [63].

A variety of protein- or peptide-based systems aimed at targeted subcellular delivery are exemplified in Table 2.

## 3. Extracellular Obstacles to Intracellular Targets during Systemic Drug Administration

### 3.1. Extracellular Barriers: Interaction with Blood Proteins and Cells

The increase in the size of drugs or drug nanoformulations compared to small molecules makes it possible to use more of the embedded necessary functions and also dramatically expands the range of substances that could be used for treatment. Currently, a huge number of such drugs are under development, including proteins and polypeptides, polymers, structures based on nucleic acids, and pharmaceutical carriers carrying many small-sized drug molecules. Their common feature is their size in the nanometer range, and all of them face similar problems on the way to their locus in quo. Any drug administered systemically into the bloodstream must overcome several natural barriers to reach the cell interior (Figure 2). These barriers matter not only for macromolecules and nanoparticles but also for many low-molecular substances that, due to the charge, cannot move across biological membranes without specialized transmembrane channels or transporters for them. The distribution of macromolecules and, to an even greater extent, nanoparticles in the body cannot be considered only in terms of physical diffusion and convection. Their interaction with blood components, passage through the walls of blood vessels and further penetration into tissues encounters a complexly regulated system for maintaining the homeostasis of the body. In this sense, the division of particles into macromolecules and nanoparticles by size is largely arbitrary. The range of sizes of nanoparticles is most often considered between 1 and 100 nm [74,75]. The average size of globular proteins usually does not exceed a dozen nanometers, while elongated proteins such as fibrinogen can have a length of up to 50 nm or more [76]. In terms of size, proteins and nanoparticles are in close ranges and follow the same patterns. Everything with a size of less than 5–6 nm (about 60 kDa for globular proteins) is excreted through the kidneys [77]. The size range of molecules and nanoparticles that can circulate in the blood for a sufficiently long time also has an upper limit. A great increase in size leads to an increase in blood clearance due to the action of the mononuclear phagocyte system (MPS), which serves to protect the body from infection [78]. The larger the particle size, the more efficiently they are absorbed by macrophages. The efficiency of this uptake depends not only on the size but also on the type of nanoparticles, their surface charge, their shape, and the properties of their surface. The nanoparticle uptake by macrophages shows an exponential dependence on their size in the range from 30 to 300 nm [79]. Incorporation of a low molecular weight drug into nanoparticles changes the drug’s pharmacokinetics, increasing its half-life time in the blood. However, along with a lifetime increase, this can lead to accumulation in undesired organs, tissues or cell types. In the case of the insufficient biodegradability of the nanoparticle base, this undesired accumulation can lead to adverse effects due to the toxicity of the nanoparticles themselves. From this point of view, the biodegradability of proteins is an advantage if they can reach their targets and do not cause long-term toxicity.

The interaction with blood proteins is of great importance for the substance injected into the blood [80]. The most abundant blood protein, albumin, is capable of binding a number of substances and possesses a long circulation period in the bloodstream with a half-life of 19 days [81]. This is provided by a special mechanism returning albumin to the blood from the endosomes using the neonatal Fc receptor (FcRn). This receptor has an increased affinity for albumin in the slightly acidic environment of the endosomes and directs it back to the cell surface. A similar mechanism also works to regulate the IgG levels [82]. Many other blood proteins are designed to remove unwanted foreign molecules and particles from the bloodstream and neutralize them. The interaction of a number of proteins with the particle surface leads to their opsonization and uptake by cells of the MPS [83,84,85]. 

### 3.2. Penetration through the Walls of Blood Vessels and Extracellular Matrix

In addition to the interaction with proteins, the walls of blood vessels, which serve to separate blood from other organs, are a natural barrier on the way of intravenously injected substances to other tissues. This barrier becomes especially significant if the drug should act on deeper lying tissue cells. The physiological upper limit of the pore size for transvascular flow through the capillary walls of blood capillaries varies for different types of tissues [86]. This limit is less than 1 nm for capillaries in the central nervous system and from 5 to 12 nm in most other tissues. Peculiarities of the vessels of the liver, myeloid bone marrow and spleen allow the extravasation of significantly large particles [86]. Many tumors are characterized by the defective architecture of the vessels and lymphatic system, as characterized by leaky vasculature and defective lymphatic drainage. For a number of therapeutic macromolecules and nanoparticles, this leads to enhanced permeability and retention (EPR) [87,88], resulting in their tumoritropic accumulation. It is believed that the EPR effect in some cases can be used to create effective anti-cancer drugs [89]. Nevertheless, the heterogeneity of real human tumors and the presence of barriers other than vascular permeability on the way to a malignant cell do not allow in most cases reliance only on EPR when creating new drugs [90,91]. The sufficiently large size of nanoparticles makes them an object of interest for cells professionally engaged in phagocytosis. In this case, even nanoparticles that have reached the tumor are absorbed mainly not by the target cells but by the tumor-associated macrophages. Even the addition of a ligand for a receptor overexpressed in cancer cells, for example, ERBB2, may not be able to correct this [92]. In addition, recent estimates show that most nanoparticles seem to penetrate across the endothelium into tumors, not through the passive EPR effect but rather via active transport processes [93].

Vascular endothelial cells form a complexly regulated system designed to ensure the transcytosis of antibodies and other blood proteins, such as albumin [94]. Transcytosis, which plays an important role in maintaining tissue homeostasis, is mediated by the caveolae of endothelial cells. For example, albumin, low-density lipoproteins, insulin, and even glucose are transported in this way [95,96]. In addition, vascular permeability is regulated by cell–cell adherens junctions that regulate tissue–fluid homeostasis, transport of nutrients and regulatory proteins, and migration of cells through the vessel epithelial barrier [97,98]. In the central nervous system, where endothelial cells are characterized by unusually highly restricted molecular exchange and very tight junctions forming the blood–brain barrier, the delivery of therapeutics becomes even more challenging. Receptor-mediated transcytosis relying on binding to specific receptors, e.g., transferrin receptors or insulin receptors, is supposed to be a prospective approach to overcome this issue [99].

The extracellular matrix, with a network of fibrillar proteins and negatively charged proteoglycans, may be another barrier on the way of some macromolecules to target cells that are not exposed to the bloodstream [100,101]. The extracellular matrix traps both positively charged and negatively charged macromolecules, so uncharged hydrophilic molecules have an advantage in overcoming this barrier [101,102]. Cells are connected to the extracellular matrix by many contacts, and the proteins and other polymers of the extracellular matrix are produced by cells. In some cases, for example, for the epithelium, the layer of glycoproteins closest to the cell, the glycocalyx, can be considered an additional barrier for macromolecules interacting with the cell [103]. The high density of the extracellular matrix, a hallmark of many malignant tumors, additionally restricts the drug trafficking in solid tumors [104]. The binding site barrier caused by the high-affinity binding of rather high molecular weight therapeutics (e.g., antibodies) at the immediate site of their extravasation area can be regarded as another barrier to the deep-tissue penetration of targeted protein-based therapeutics [105].

## 4. Intracellular Targets and Barriers on the Way to Them

When the drug finally overcomes the aforementioned barriers on its way to the cell, further aspects regarding its direct impact on intracellular targets arise. For an accurate effect on the intracellular target, it is necessary to take into account at least two interrelated issues: the intracellular location of the target and what else can be changed as a result of the drug action. Moreover, upon the development of such drugs, one should take into account their fate after cell entry, including intracellular processing and degradation. A huge amount of experimental material has been accumulated on the intracellular distribution of proteins and other macromolecules as well as their effects on intracellular processes. The systematization of these data and the transfer of high-throughput proteome analysis methods to the subcellular level have begun to attract close attention [106].

A living cell, like a living organism, is a structured system divided by phospholipid membranes into compartments. The penetration of macromolecules into the cell is a strictly regulated process (Figure 3). The simple penetration of macromolecules into the cell in most cases does not provide interaction with its molecular target, as the simultaneous presence of their target in the same compartment is required. The plasma membrane is the first of the cell barriers to intracellular targets. The natural way of penetration of macromolecules into the cell is receptor-mediated endocytosis [107,108]. This pathway depends on the type of cell and the receptors expressed on the cell surface. There are quite a few types of uptake of macromolecules and particles by cells. These include the clathrin-, caveolin-, and endophilin-mediated types of endocytosis that are dependent on dynamin, as well as the dynamin-independent ARF6- and Cdc42-mediated and other types of endocytosis, macropinocytosis and phagocytosis [107,109,110,111,112]. The pathways of endocytosis vary greatly, both in kinetic characteristics and in the size of the vesicles formed. A number of proteins that are involved in the machinery of vesicular transport from the plasma membrane can participate in several variants of endocytosis. They form a complementary network of transport routes adapted to the tasks of a particular type of cell. From the point of view of reaching intracellular targets, it is important that most of the intracellular targets remain inaccessible to the drug after its uptake by endocytosis. As a result of all types of endocytosis, areas of the plasma membrane are closed around receptor complexes with ligands attached to them. After that, the resulting closed membrane formations (vesicles or larger phagosomes) move inside the cell. This movement does not lead to the penetration of macromolecules bound to their receptors through the phospholipid membrane. Macromolecules entering cells in this way can be transported to lysosomes and returned back to the plasma membrane or can be transported to other membrane formations separated from the rest of the cell. During endocytosis, receptors and ligands are sorted based on the presence of special signals and changes in the properties of the endosome medium, primarily its acidification. This determines whether the endosome content will be recycled or degraded [113,114,115].

The interaction and exchange of contents between the endosomes and other membrane organelles, as well as with the plasma membrane, are carried out by a vesicular transport system [116,117]. These organelles include the Golgi complex (Golgi apparatus) with the trans-Golgi network (TGN), ER, peroxisomes and lysosomes.

There are also other organelles in the cell, the nucleus or mitochondria, surrounded by two layers of membranes into which proteins are transported only from the cytosol but not from other membrane organelles [118,119]. A large number of different NLS that mediate the entry of proteins via the nuclear pore complex (NPS) into the cell nucleus are well known [120,121,122,123,124]. Mitochondrial transport is one of the most prominent routes for the intracellular transport of macromolecules, since only 13 of the approximately 1500 mitochondrial proteins are encoded by mitochondrial DNA. There are also several ways for the transport of proteins into the mitochondria, which direct proteins into the inner space of the mitochondria, into the intermembrane space, as well as into the inner and outer membranes of the mitochondria [125,126]. The characteristics of those sequences that are recognized by the translocase of the outer membrane of mitochondria (TOM) and translocase of the inner membrane of mitochondria (TIM, mainly TIM23) complexes for transport into the mitochondria are also known for many proteins [127,128,129,130].

An in-depth description of the pathways for the targeting of each individual intracellular organelle is provided in Section 8.

In addition to the membrane intracellular barriers, the movement of macromolecules to the therapeutic target can also be hindered by the intracellular environment itself due to the large concentration of macromolecules and the presence of organelles in the cell, which complicate diffusion. An extremely high concentration of proteins in the cell leads to the effect of macromolecular crowding, which is caused by a sharp reduction in the solvent volume available for moving macromolecules, with an increase in their concentration [131,132]. In cells, this effect is enhanced due to the high proportion of non-spherical macromolecules [133]. For many substances, such a description of the movement inside cells is simplified due to the formation of biomolecular condensates that form separate (sub)compartments in the cytosol and nucleoplasm [134]. These biomolecular condensates can be considered different phases in liquid–liquid separation [135]. In this regard, the possibility to assess the effective concentration of the delivered drugs acting on selected regulatory pathways can be very helpful for their further successful development [136].

## 5. Macromolecular Delivery Systems

### 5.1. Properties of Macromolecular Delivery Systems Predetermined by Extracellular Barriers

There is no single list of barriers that a macromolecular agent must overcome in order to reach its target. This list depends on the intracellular location of the target, on what type of cells the target is located in and where those cells are located in the body. Accordingly, there is no single solution to the problem of drug delivery into the cell. It is also clear that the properties of the delivery system may also depend on the drug being delivered. Particles close in size to such typical pathogens as viruses and bacteria, due to their size, are convenient means for the delivery to macrophages and other MPS cells and probably to some tumors due to the EPR effect. Their advantages include the possibility of the delivery and subsequent local release of a large number of low molecular weight drugs with an already known effect. For delivery inside most other types of cells, especially those lying deep in the tissue, it is more preferable to use something smaller. The lower limit of the size of such delivery systems is the aforementioned renal filtration limit. Quite large proteins exceeding ca. 60 kDa fall into this range. The most well-known solution to the problem of the renal excretion of insufficiently large molecules is the addition of polyethylene glycol [137]. A long hydrophilic PEG residue shielding macromolecules or nanoparticles from interaction with MPS can also reduce the effectiveness of their action, sterically hindering the interactions necessary for their functioning. In addition, as shown by the results of the clinical use of PEGylated drugs, PEG can also cause the formation of antibodies to itself [138,139]. There are alternative variants of protein modifications that increase their lifetime in the bloodstream. It is noteworthy that these modifications are sequences consisting of proteinogenic amino acids. These include the large unstructured polypeptide XTEN [140,141], which includes serine, glycine, proline, threonine, glutamic acid and alanine, as well as polypeptides based on proline, alanine and serine (PAS) [142], or elastin-like polypeptides (ELP) with a pentapeptide repeat—Val-Pro-Gly-Xaa-Gly (VPGXG)—found in elastin [143].

### 5.2. Properties of Macromolecular Delivery Systems Predetermined by Active Principles

Recognition of the cell type, transport of macromolecules into cells, and signal transduction via intracellular pathways are all largely carried out through the interaction or modification of proteins. The variety of amino acid sequences makes it possible to combine the necessary transport functions with therapeutic ones within a single molecule encoded by a single gene [68,144,145]. This makes it possible to easily obtain a reproducible product and ensure the scalability of its production. For this kind of drug system, it is necessary to preserve the properties that ensure both transport to the desired part of the target cell and the performance of the delivered drug.

The use of artificial polypeptides as a delivery system imposes its own natural limitations associated with the attachment of the active principle. Larger nanoparticles can be loaded with a large number of low molecular weight drugs. The payload of protein molecules with low molecular weight drugs is usually inferior to relatively large nanoparticles, especially liposomes. The attachment of the active principle may also require the presence of a degradable linker for its therapeutic effect. An obvious solution to increase the effect of the delivered substance is the use of a component that provides the production of a large amount of active principle. Therefore, a delivered drug that is capable of generating a greater number of active products upon an external stimulus or on its own is attractive for creating effective macromolecular drug delivery systems. These include, for example, photosensitizers that produce a variety of reactive oxygen species (ROS) when irradiated with light of the appropriate wavelength. These ROS are extremely reactive, so they are not able to spread over a long range in biological media and can damage biological molecules in the immediate vicinity of the emission site [146]. Another variant of such an active principle is the emitters of Auger electrons, the range of which, as a rule, is limited to a few tens of nanometers [17]. These variants of the active principle can be considered an example of locally acting agents that need targeted intracellular delivery to realize their effect [147].

The advantage of proteins is the possibility of including the necessary regulatory sequence or a fragment with the necessary enzymatic activity directly into the delivery system as part of a single amino acid sequence. Examples of such protein drugs are immunotoxins for the treatment of cancer. They are fused proteins that include catalytic subunits of natural toxins, for example, ADP-ribosylating pseudomonas exotoxin and diphtheria toxin, or a ligand to the receptors of malignant cells, which can be presented by both an artificial antibody and a natural ligand [27]. Three immunotoxins have already been approved for the treatment of hematological malignancies: IL-2 fused to a diphtheria toxin fragment (Denileukin diftitox) for the treatment of T-cell lymphoma, anti-CD22 antibody fused to a pseudomonas exotoxin (Moxetumomab pasudotox) for the treatment of hairy-cell leukemia, and IL-3 fused to a fragment of diphtheria toxin (Tagraxofusp) for the treatment of blastic plasmacytoid dendritic cell neoplasm [27]. Unfortunately, the extremely high toxicity of ADP-ribosylating enzymes, which are the active principle of such systems, will hinder their widespread use. Nevertheless, these examples demonstrate the possibility of successfully overcoming the barriers on the way from the bloodstream to the cytosol of a target cell, at least for the circulating cells. However, it is clear that it will be significantly more difficult to reach the cells located deep in the tissues.

### 5.3. Combining Transport Functions in a Single Polypeptide Chain

The targeted intracellular delivery to cells of a specified type using macromolecules requires combining many transport functions in a single molecule. In general, this is a non-trivial task; however, the already created molecules have demonstrated that its solution is quite possible (Figure 4). 

The idea of using the combination of several amino acid sequences and transport modules for the delivery of a drug into the nuclei of target cells was originated by A.S. Sobolev and first published in 2000 [148]. The development of this approach has shown that the delivery of locally acting cytotoxic agents such as photosensitizers [149,150], α-emitters [151] and Auger electron emitters [152,153,154] greatly increases their effectiveness in destroying cancer target cells. In some cases, this increase exceeds three orders of magnitude [149,154]. One of the key features of this approach is the modular principle of their construction: a certain sequence or module is responsible for each transport function. Therefore, such artificial protein structures are called modular nanotransporters (MNTs) [68,155]. Modules can be removed, added, replaced or rearranged. The use of MNTs for photodynamic therapy or for Auger electron emitter-based radionuclide therapy of experimental mouse tumors has demonstrated the significant suppression of tumor growth and cure of animals [45,46,153,156,157]. These results were achieved with MNTs consisting of four functional modules. The first of them is a ligand module that binds to an overexpressed internalized receptor on the surface of cancer cells. This module provides receptor-mediated endocytosis into the target cell. The second module, the endosomolytic one, provides a pH-dependent escape from the endosomes in slightly acidic conditions. The third module includes the nuclear localization sequence and provides the transport of the MNTs into the cell nucleus as a result of the interaction with the α/β-importin complex. The fourth module separates the modules from each other, which is necessary for their effective functioning. This module also combines the other modules into a single unit and, therefore, is a carrier module. There are countless implementations of such constructions, and there is a long way to go to improve them until the optimal variants for point impact on each intracellular target will be developed. Currently, only some of the conditions that should lead to success on this path are fully clear. In particular, it is possible to evaluate those conditions that allow the creation of structures for the effective delivery of therapeutic agents to the target cell nucleus. The key points for targeted intracellular drug delivery are recognition of the required cell type, endocytosis in them and redirection to the compartment where the target is located.

Moreover, tumor recognition specificity can be further enhanced by the inclusion of masking domains for the selective unmasking of cell targeting domains by tumor-specific proteases cleavage [158,159,160].

## 6. Protein-Based Ligands for Intracellular Delivery

As noted above, there is a large class of cell surface receptor proteins whose main function is cell regulation. These receptors are able to bind natural ligands and undergo endocytosis. There are quite a few successful drugs that are natural ligands to internalizable receptors. Their artificial analogues are also often used. Many of these drugs, such as insulin, ACTH, and somatotropin derivatives, have become widespread [161]. In addition, there are currently standard technologies for the creation of monoclonal antibodies and antibody mimetics [13,162,163], which are widely used to affect these receptors. In 2021, the hundredth mAb received FDA approval for treatment [164], and currently, the number of antibodies approved for clinical use exceeds 130 [165]. Despite significant costs in development and production, this class of drugs shows more frequent success in moving from phase 1 clinical trials to approval than small molecules [166]. Many of these antibodies have proven to be in high demand and have more than a billion-dollar market. For example, seven of the ten best-selling drugs in the world in 2019 were antibodies [167]. This indicates a great potential for the use of artificial polypeptide drugs, which is likely to be increasingly revealed in the foreseeable future.

Despite this apparent success, the antibodies that are already approved and being tested cover only a small fraction of possible targets for disease treatment. The technologies used in their creation include the production of chimerical polypeptides and can be used to solve a much wider range of tasks. There are a huge number of therapeutic targets inside cells that are undruggable for small molecules [168], so intracellular delivery of antibodies [18] and transcription factors [169] is attracting increasing attention. Along with antibodies, their Fab fragments, camelid single-domain antibodies, and antibody mimetics based on various scaffolds can be used to reach these undruggable targets. These include, for example, affibody based on the Z-domain of staphylococcal protein A [170], anticalins derived from human lipocalins [171], DARPins (designed on the basis of repeats of the sequence from ankyrin) [172], knottins (short knot sequences) [173], monobodies (adnectins) based on a fibronectin type III domain [174], and others [163,175]. All these variants are significantly smaller in size than IgG and can be included by standard methods of genetic engineering in a polypeptide molecule carrying additional functions.

## 7. Use of Receptor-Mediated Endocytosis for Cell-Specific Drug Delivery

From the point of view of drug delivery into a specified cell type, the use of receptors has several obvious consequences that depend on the type of ligand chosen. When using natural ligands to receptors for drug delivery into the cell by endocytosis, in addition to the uptake of ligand–receptor complexes, receptor-mediated regulation of cellular processes takes place. This can lead to side effects in cells that narrow the therapeutic window. Artificial ligands created on the basis of antibodies and antibody mimetics, as a rule, are antagonists for receptors. This leads to the blocking of the process activated by the natural agonist. This blocking can also cause side effects. A typical example is the side effects that occur during the treatment of malignancies by blocking the tyrosine kinase receptors [176,177]. In addition, receptor blockers, as a rule, do not undergo endocytosis or do it significantly slower than activated natural ligands. The average lifetime of proteins in the plasma membrane is estimated in tens of hours [178]; therefore, even in the absence of a signal mediating endocytosis, ligands attached to surface proteins can slowly penetrate into cells during membrane renewal due to the exchange of its parts with intracellular organelles. It is not clear, however, whether slow delivery and, accordingly, a small fraction of the delivered substance make it possible to achieve a sufficient therapeutic effect for the intracellular concentration of the drug. For some receptors that undergo ligand-induced endocytosis, there is also parallel constitutive ligand-independent internalization. For example, this was shown for the epidermal growth factor receptor (EGFR) [179]. This system, apparently, serves to simultaneously provide both high sensitivity and stability to the signal transmission system via the EGFR. It should also be mentioned that the types of endocytosis can act at significantly different rates, which provides both the possibility of a rapid response and additional regulation.

As was mentioned above, a number of uptake ways of macromolecules and nanoparticles are known [108,111,112]. However, not all of them are present in all cell types [108]. The most well-known example is phagocytosis, which is predominantly used by MPS cells. Another example that highlights the importance of differences between cells in endocytosis types is a comparison of the brain vessel endothelium with the endothelium of peripheral tissues. In the first of them, caveolae are practically absent, whereas in the second, their number is large, ensuring the uptake and transcytosis of a number of substances [180]. The type of endocytosis also depends on the types of receptors expressed on the cell surface [181]. Meanwhile, the same receptor can be internalized in different ways depending on the type of cell, the ligand used, and the concentration of the ligand. A part of the key internalization proteins is shared by several different pathways of receptor-mediated endocytosis, resulting in their crosstalk [182]. Apparently, in general, endocytosis can be represented as a regulated system that is adapted to the tasks of a specific cell type. Depending on the conditions, the internalization of the receptor may switch from one type of endocytosis to another. Most of the data on the mechanisms of endocytosis were obtained on cell cultures, so the real mechanisms of uptake of drug delivery systems in vivo are often unknown.

## 8. Branching Delivery Pathways for Endocytosed Drugs

### 8.1. Endosomes Acidification and Protein Drug Delivery

As already mentioned, the endocytosed ligand–receptor complex, when passing through the initial stages of the endocytic pathway, undergoes sorting (Figure 5). This pathway is a highly dynamic endocytic membrane system, which is comprised of early endosomes, recycling endosomes, late endosomes/multivesicular bodies, and lysosomes. The main pathways followed by internalized macromolecules in the majority of cells are the pathway to the lysosomes for degradation and recycling back to the plasma membrane, which involves the reuse of receptors [114]. An additional variety comes from the exchange with the TGN and, through it, with the ER. These pathways are largely determined by the receptors used and the ligands chosen. To achieve different intracellular compartments, drug delivery systems must use additional components. The most commonly used intracellular drug targets are the nucleus, cytosol, and mitochondria. Therefore, it is not surprising that much attention is paid to the study of the possibility of drug release from endosomes into the cytosol [183,184,185,186,187,188,189]. Namely, the cytosol is the cell part from which transport into the nucleus and mitochondria is possible. To deliver drugs to the cytosol, the delivery system must be able to leave endosomes before entering lysosomes. Otherwise, the delivered drugs are subject to degradation by lysosomal hydrolases. The acidification of the endosomal content, which takes place when molecules move along the endocytic pathway, is often used to deliver internalized substances to the cytosol [190,191,192,193,194]. There are rather many amino acid sequences that can provide a disturbance of the lipid bilayer integrity in a slightly acidic environment. Many of them are used to study the possibility of effective delivery of drugs to the cytosol [195]. For drug delivery into the cytosol of a target cell by receptor-mediated endocytosis, it is critical that the interaction of such a sequence with the lipid bilayer and pore formation or membrane rupture occur in a slightly acidic environment at pH 6.5–5.5. The peptide sequence interacting with the phospholipid bilayer under neutral conditions will undergo nonspecific embedding into the plasma membrane of any cell. It is clear that the specificity of drug delivery to a specified cell type is hardly achievable in this case. At the same time, if such a sequence disrupts the lipid bilayer at pH 5 or below, the delivered drug will be hydrolyzed in lysosomes [196].

For many years, a promising research direction that uses CPP to deliver drugs into the cytosol has been developing [197,198,199]. The term CPP is used to refer to a group of peptides that can cross phospholipid membranes and reach the cytosol. This group includes several different types of peptide sequences, which differ greatly in their composition as well as in their known and proposed mechanisms of action [200]. Most of them are cationic sequences that easily bind to the anionic surface of any cell. Such properties are poorly compatible with delivery into a specified cell type.

In addition, most CPPs work effectively only at micromolar concentrations, which are significantly higher than the dissociation constants of the ligand–receptor complexes of cell surface receptors. Therefore, when using concentrations optimal for specific receptor-mediated delivery, it is difficult to ensure an effective exit from endosomes using such sequences. It should also be taken into account that the delivery of macromolecules to cells has been shown only for an extremely small (about 4%) part of the sequences proposed as CPPs [200]. Research on other CPPs has been limited to the transportation of fluorescent dyes, small molecules, and short peptides.

Another option for delivery to the cytosol is to use fragments of a number of bacterial toxins that are specialized for this process (Figure 5). They include *Corynebacterium diphtheriae* toxin [201], anthrax toxins [202], *Clostridium botulinum* C2 toxin [203], binary toxin (CDT), toxin A (TcdA) and B (TcdB) from *Clostridium difficile* [204,205], and others [206,207,208]. An essential feature of these toxins is their ability to undergo conformational rearrangements in a weakly acidic environment. The protonation of the anionic groups of these proteins with a decrease in the environmental pH leads to a decrease in their charge and an increase in their hydrophobicity, with subsequent embedding in the bilayer. These toxins and the chimeric polypeptide constructs based on them are capable of forming pores in artificial phospholipid membranes [209,210,211,212,213]. These pores can be seen on flat bilayers using atomic force microscopy [211]. Nevertheless, it has been found that such toxins additionally use the cellular proteins to increase the efficiency of delivering their cargo [201,214]. The use of bacterial toxins’ ability to transport proteins into the cytosol is considered a suitable platform for drug delivery [186,215,216]. The advantages of using the translocation domains of toxins for drug delivery into the cytosol include their natural suitability for the transport of sufficiently large proteins. This greatly expands the range of possible active agents, including proteins that compete for target proteins that transmit regulatory signals within the cell. Short CPPs or anionic amphipathic peptides such as GALA [217] may be sterically hampered when delivering large cargo across membranes.

### 8.2. Drug Delivery into the Nucleus

As noted above, drug delivery to the cytosol also opens the way to molecular targets in the nucleus and mitochondria that are attractive for therapy. The nucleus is the main object for regulating gene expression, both for disease correction and for in vivo cell reprogramming. The endosomes’ escape is necessary, but it is not sufficient for the transport of drugs into the cell nucleus from the cytosol. The transport of macromolecules from the cytosol into the nucleus is provided by the NPC [218]. The NPC is an effective barrier for the passive diffusion of macromolecules, highly limiting their traffic across the NPC with the increase in their size [219]. Efficient transport into the nucleus of macromolecules with a mass of greater than 40 kDa is mediated by importins, proteins that bind proteins with the NLS [120,121,122,123,124]. The NLS are an attractive means of transport into the nucleus, which is often used to increase the efficiency of the proposed intracellular drug delivery systems. Various NLS are often included in constructs based on nanoparticles, polymers, micelles, and various polypeptides, including natural proteins, antibodies and their derivatives, and synthetic peptides for the targeted delivery of the active agent into the nucleus [19,45,46,66,68,149,150,152,153,154,155,156,157,193,220,221,222,223,224,225,226]. In addition, for the successful transport of the drug into the nucleus, it is necessary to take into account the peculiarities of macromolecules’ diffusion in the cytosol and the mechanisms of their active movement. The movement in the cytosol is limited by molecular crowding as well as obstructions due to the cytoskeleton and organelles [132,227]. Active transport through the cytosol is carried out along microtubules by means of special motor proteins that transport macromolecules toward the centromere near the nucleus (dynein) or toward the cell periphery (kinesin) [228]. The inclusion of a dynein light chain or sequences that bind to dynein in a drug delivery vehicle can increase its effectiveness [229,230].

### 8.3. Mitochondrial Drug Delivery

The mitochondria are also a cellular compartment that is accessible to the drug only from the cytosol (Figure 5). Studies of the single-cell proteome show that the proportion of “tissue-rich” proteins in the mitochondria is apparently small [30], which underlines the universal importance of the uniform functioning of these organelles for cells. Most mitochondrial proteins possess a mitochondrial targeting sequence (MTS). About half of these MTSs are localized at their N-terminus, which is probably explained by the close connection of their translation near the mitochondria [231]. Mitochondrial proteins do not have conserved MTSs; however, some possess a propensity to form an amphiphilic helix with one positively charged surface and one hydrophobic surface [232]. There are different types of MTSs for proteins that are sent to the outer and inner mitochondrial membranes, the intermembrane space, and the mitochondrial matrix [233,234,235,236,237]. Proteins are transported mainly via the TOM complex located in the outer membrane and the TIM complexes located in the inner membrane. The narrow channels of the TOM and TIM complexes allow the transport of proteins only in the unfolded state, which is limited to structures no thicker than an alpha helix. Therefore, generally, transport into the mitochondria is closely associated with the activity of chaperones and regulated by them [237,238]. A significant part of the MTSs is cleavable [236,237]. The use of MTSs is considered one of the perspective directions for the development of drug delivery systems into the mitochondria [16,239,240,241,242,243]. MTSs generally do not possess tissue specificity, which makes it possible to include MTS in various drug delivery systems.

### 8.4. Peroxisomal Drug Delivery

Another cellular compartment that receives proteins from the cytosol is the peroxisome (Figure 5). Peroxisomal proteins can be transported into these organelles in both folded and partially unfolded forms. A number of peroxisome targeting signals that provide binding to cytosolic receptor proteins for co-transport into peroxisomes are known [244,245]. Part of the peroxisome membrane proteins come from the ER via vesicular transport [244,245]. The delivery of enzymes to peroxisomes is considered a useful tool for the treatment of a number of diseases associated with impaired oxidative processes [246,247].

### 8.5. Vesicular Transport to Intracellular Targets

Delivery to other intracellular compartments can be provided by vesicular transport from endosomes (Figure 5). As already discussed above, one of the main pathways of endocytosis ends in lysosomes. This pathway can also be used for drug delivery for the treatment of the corresponding diseases. For example, this approach is being developed for lysosome enzyme delivery for the treatment of lysosomal storage diseases. [248]. There is also a specialized pathway for protein delivery from the plasma membrane into lysosomes through mannose-6-phosphate receptors [249].

One of the possible ways of intracellular vesicular trafficking is the protein exchange between the endosomes and the TGN [250,251,252]. This exchange has been detected for a number of essential proteins, including the mannose-6-phosphate receptor [253], vacuolar protein sorting-10 (Vps10) proteins [254], and furin [255]. Some plant and bacterial toxins, for example, ricin [256], Shiga toxin [257], and cholera toxin [258], are transported via this pathway. The Shiga and cholera toxins belong to the AB5 family of toxins, whose b-subunit provides transport to the TGN and further steps of intracellular transport, including moving into the ER and cytosol [259]. The b-subunits of these toxins are considered potentially nontoxic carriers for drug delivery [260]. The TGN is part of the pathway for protein trafficking, modification, and sorting, which provides both the protein secretion and the return of the necessary transport components back to the Golgi complex and to the rough ER [261,262]. The coordinated functioning of all the parts of the Golgi complex is necessary for the normal functioning of the cell [261,263]. Special signal sequences ensure the retention of proteins in the Golgi complex [262,264,265,266]. It is also possible that the transport of some proteins from the Golgi complex is affected by their polymerization. The disassembly of their polymer complexes facilitates the return back to the Golgi cisternae (the ER–Golgi intermediate compartment, ERGIC) [267]. The Golgi complex is considered a promising target in the treatment of cancer, fibrosis, amyotrophic lateral sclerosis, Creutzfeldt-Jacob disease, and other neurological diseases [260].

The Golgi complex constantly exchanges content with the ER via vesicular transport. The short KDEL sequence provides a dynamic exchange of many proteins between the ER and Golgi [268]. The KDEL receptor binds these proteins in the ER lumen at pH 7.2–7.4 and releases them at pH 6.2 in the cis-Golgi. The ER is the largest compartment of the cytoplasm and can occupy more than a third of its volume [269]. Processes such as the ER stress response and the unfolded protein response (UPR) are of interest for delivery into the ER, both for destroying cancer cells and for treating diseases.

There is also a channel between the ER and the cytosol through which cellular proteins can be transported from one compartment to another. For transport from the cytosol to the ER, it works mainly for newly synthesized proteins that are designed to function in the ER or outside the cell. This transport is mediated by the Sec61 translocon, which interacts with the ribosomes of the rough ER. Thousands of secretory and membrane proteins follow this pathway [270,271]. The Sec61 translocon likely also works in the opposite direction for ER-associated protein degradation (ERAD) by transporting misfolded proteins into the cytosol [272,273]. Apparently, there are several retrotranslocons that provide the reverse transport of misfolded and unfolded proteins from the ER to the cytosol, including Sec61 [273], the Hrd1–Hrd3 complex [274], Derlin family members (Der1-3) [275], and the Hrd1–Der1 complex [276]. The process is closely related to the ubiquitination and degradation of such proteins in proteasomes [277,278]. In addition, Sec61 seems to mediate the transport of a number of receptors after the internalization of ligand–receptor complexes and the vesicular transport of some of them from the endosomes to the TGN and further into the cytosol and cell nucleus. Such a process has been noted for one of the most frequently overexpressed receptors in cancer, the epidermal growth factor receptor (EGFR) [279], and for some other tyrosine kinases [280]. The throughput of this additional retrograde transport pathway of endocytosed macromolecules is much inferior to transport into lysosomes or recycling to the cell surface. The presence of this macromolecule transport pathway is necessary for some bacterial toxins, which, after leaving the ER, are folded in the cytosol and not cleaved by the proteasome [257,281,282,283].

The vast majority of human proteins enter the lumen of the ER during translation. However, about 200 proteins are known, mostly small and mostly containing a hydrophobic site at the c-terminus, which are translocated post-translationally [284]. Thus, in principle, proteins from outside the cell can deliver medicine into the ER in two main ways. One of them passes through endosomes and subsequent retrograde transport through the Golgi complex, as described above for drug delivery systems based on AB5 toxins [285]. Another group of toxins, which includes diphtheria toxin, provides the means to release the delivered drugs directly from the endosomes [150,286]. Therefore, the second way may be provided via endosome escape with subsequent translocation into the ER.

## 9. Future Prospects

The development of drug delivery systems that control the interaction between a drug and a biological microenvironment is one of the main current paradigms for the creation of new pharmaceutical products [287]. Proteins are the main class of substances in the body that both respond to and make the cell microenvironment. The available variety of interacting proteins, together with the membrane barriers, largely determines the distribution of drugs inside the body and the responses to them. Disturbance of the interactions of proteins with each other and with other cellular structures is the cause of many pathologies. Blocking aberrant intracellular interactions and restoring disturbed regulatory pathways could serve as the basis for the next level of truly personalized medicine. This promising area is able to open new horizons for the effective use of a cohort of already approved therapeutics, whose targets are now limited to cell surface receptors. The analysis of the currently available data and the emerging prospects allow us to conclude that a number of tasks need to be solved for success in this area. The successes achieved in the last decade allow us to hope for their solution in the foreseeable future. The development of transcriptomics and proteomics for the whole body and single cells provides a basis for understanding what variants of delivery systems can be used for treatment and what side effects can be expected when using them.

Proteins offer the unique opportunity to combine several biological functions within a relatively small structure. An example of this kind of artificial molecule is modular nanotransporters that combine several transport functions: binding to the receptor and endocytosis, endosome escape and transport into the cell nucleus [45,46,68,149,150,153,154,155,156,157,193,224]. Those artificial protein molecules combining several polypeptide moieties can be either produced by genetic engineering or by various post-translational protein assembly approaches (e.g., barnase-barstar-based [288], enzymatic-mediated protein ligation [289,290], intein-mediated ligation [290,291], and SpyTag/SpyCatcher [292]). The delivery of cytotoxic agents to the most vulnerable cellular compartments is the most obvious opportunity for cancer treatment. The same principle can be used to treat many regulatory disorders by targeting the delivery of the appropriate regulatory sequences into the correct compartment in the target cell. There are a number of requirements for this method of influencing disorders to be implemented. Among these are requirements related to the composition and design of the constructs, the requirement for knowledge of both the whole body and the intracellular distribution and transport of macromolecules, and an understanding of the protein–protein interaction. The status of research related to the components that are necessary for intracellular delivery constructs, as well as research on the distribution of proteins in cells, is briefly reviewed here. Several other problems are also significant in the development of delivery systems for intracellular targets in cells of a specified type. Although a lot of attention is paid to the problem of the protein structure, there is no experimental solution, even for many natural proteins [293]. This complicates the process of developing artificial polypeptide structures for transport and obtaining an effective treatment. Nevertheless, the rapid progress in the structure prediction of previously unknown protein structures [294] allows us to count on the possibility of predicting the structure of an artificial protein designed for drug transport to intracellular targets. Three-dimensional structures generated by AlphaFold have a median backbone accuracy near one angstrom [295]. Approaches that can predict the three-dimensional structure of multi-domain proteins [296] and proteins consisting of several chains [297] are also being developed. The prediction of protein structures is complicated by the presence in the proteome of a large number of intrinsically disordered proteins, the structures of which may poorly correspond to any predictions [298]. Artificial protein constructs assembled from domains with different functions can also coexist in a solution in several different conformations. This has been found, for example, for modular nanotransporters for delivering cytotoxic agents into the nuclei of cancer cells with the overexpression of the EGFR or the melanocortin 1 receptor [299]. Another problem is the incompleteness of the data on the space of intermolecular and, above all, protein–protein interactions. The available maps of the human interactome cover mainly binary interactions and do not take into account either their strength or the possible nonlinearity of these interactions [300,301]. The extraordinary complexity and lack of full data on the regulation of the body are additional barriers to the creation of constructs aimed at intracellular targets [302]. A typical way to solve this problem is to identify the main parameters that control the system. In many cases, this path leads to success. Only future research can show the usability, limitations, and optimal frames for this reductionism.

## 10. Conclusions

The performed analysis reveals that protein constructs for delivery to a certain cell compartment of a specified cell type can be considered a distinctly promising class of drugs and delivery systems for precise drug targeting. Such protein constructs can serve either as components of intensively developed drug nanoparticles or as a significant extension to the list of promising therapeutic agents. The main prerequisites for the development of this research direction are:Generation of datasets on protein expression in individual cell types;Many-fold differing expression of a significant part of the protein targets, both by cell types and by subcellular localization;A significant change of protein expression in various pathologies, primarily in cancer;Elucidation of the plethora of barriers on the way of a macromolecule to its target site after administration at the whole-body-, tissue- and cellular-level and the ways to overcome them.

Moreover, significant efforts are already being made in research in the following areas, which are of crucial importance for further movement in the aforementioned direction:Accumulation of data on the subcellular distribution of proteins in cells of various types and their individual differences;Significant development of interactome maps with its transition from ascertaining the presence of interaction between proteins to its concentration and kinetic dependences;Progress in predicting the spatial structure of the created artificial protein constructs and their interaction with the proteome.

The creation and development of protein structures for precise impacts on intracellular targets requires huge efforts and investments, which do not seem, however, overwhelming compared to the funds that humanity spends on wealth accumulation and conflicts.

## Figures and Tables

**Figure 1 pharmaceutics-15-00987-f001:**
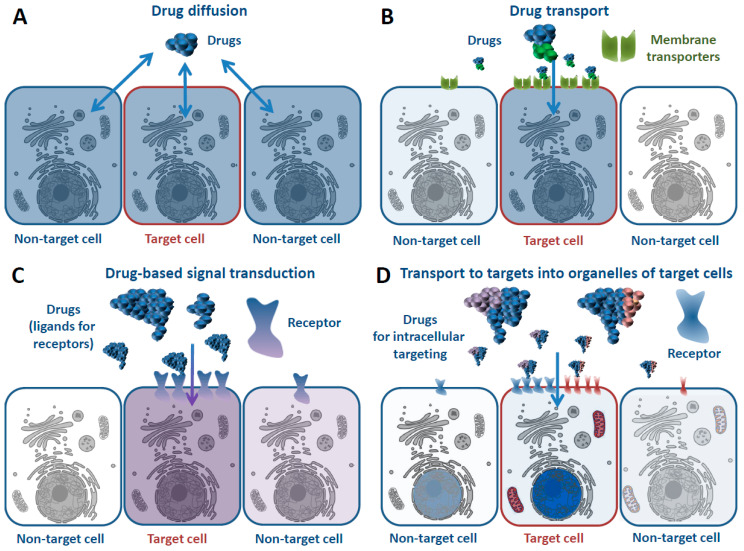
Pathways to intracellular drug targets. (**A**) Diffusion of small non-charged molecules. (**B**) Transport of drugs via transmembrane transporters. (**C**) Receptor-mediated signal transduction. (**D**) Receptor-mediated drug transport into specified cell compartments.

**Figure 2 pharmaceutics-15-00987-f002:**
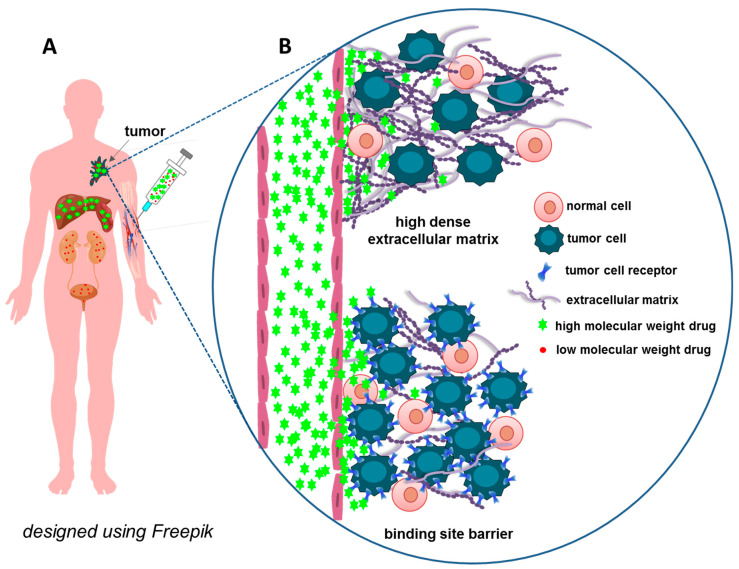
Size-dependent extracellular barriers on the way of systemically administered drugs to intracellular targets. (**A**)—Whole body level barriers. Drug formulations measuring less than 5–6 nm (about 60 kDa for globular proteins) are excreted predominantly via the kidney; larger drug formulations are mostly eliminated via the mononuclear phagocyte system, with the liver and spleen being among the largest reservoirs. (**B**)—Some examples of tissue (tumor tissue is presented here) level barriers on the pathway of a high molecular weight (e.g., protein-based) drug following its extravasation.

**Figure 3 pharmaceutics-15-00987-f003:**
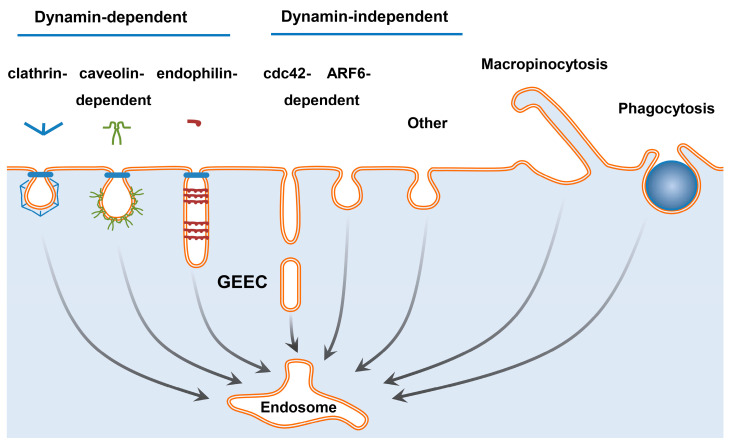
Examples of the main endocytosis pathways. GEEC—glycosylphosphatidylinositol-anchored protein-enriched compartment.

**Figure 4 pharmaceutics-15-00987-f004:**
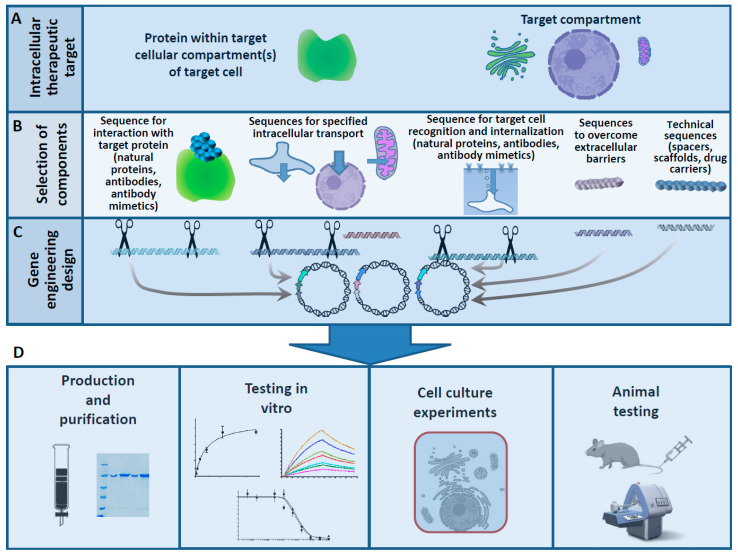
Schematic representation of the stages of the development of protein-based therapeutics for selective drug delivery into a specific compartment of target cells. (**A**) Choice of the intracellular target. (**B**) Selection of the necessary components of the developed therapeutics. (**C**) Gene engineering of the protein-based therapeutics or its part. (**D**) Production and validation of the developed therapeutics.

**Figure 5 pharmaceutics-15-00987-f005:**
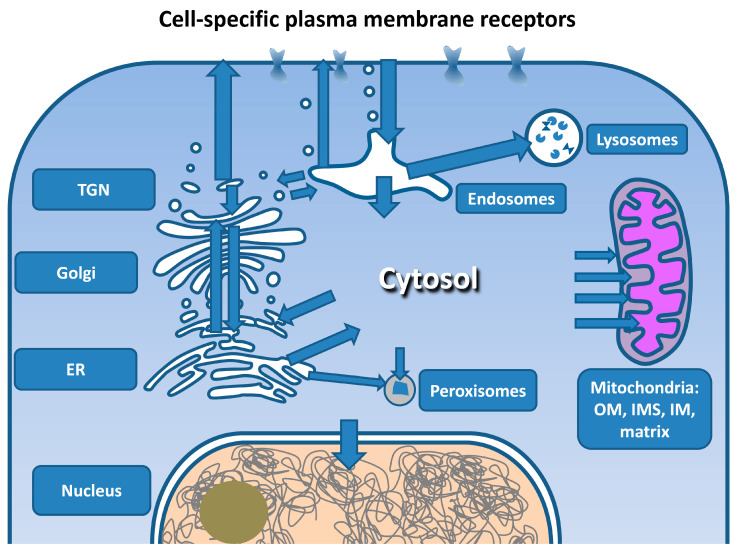
Main pathways of cell-specific macromolecular drug delivery. TGN—trans-Golgi network; ER—endoplasmic reticulum; OM—outer mitochondrial membrane; IMS—intermembrane space; IM—inner mitochondrial membrane.

**Table 1 pharmaceutics-15-00987-t001:** Estimation of the intracellular distribution of proteins, which are specific for a particular cell type, according to the single-cell RNA-seq data for genes with evidence at the protein level from the Human Protein Atlas (https://www.proteinatlas.org/, accessed on 16 March 2023).

Compartment	All	Cell Type Enriched *	Group Enriched **	Low Cell Type Specificity
Plasma membrane	1999	152	286	258
Cytosol ***	5550	330	576	1203
Nucleus ****	7285	385	725	1926
Mitochondria	1109	46	100	263
Endoplasmic reticulum	527	43	51	145
Golgi apparatus	1134	87	139	218

*—at least fourfold higher expression levels in one cell type as compared with any other analyzed cell type; **—genes with enriched expression in a small number of cell types (2–10); ***—including the cytoskeleton and the centrosome; ****—including the nucleoplasm, the nuclear membrane, the nucleoli, the nuclear bodies, the nuclear speckles, and the nucleoli fibrillar center.

**Table 2 pharmaceutics-15-00987-t002:** Protein-based systems for subcellular targeted delivery.

Delivery System	Cargo		Targeted Subcellular Compartment/Complex	Clinical Use	Refs.
	Delivered	Recruited			
Antibody drug conjugates	Wide panel of cytotoxic agents (e.g., methotrexate, doxorubicin, radionuclides, calicheamicin)		Usually lysosome, where the drug is cleaved from the conjugate	More than 10 are already used in clinical settings, e.g., trastuzumab emtansine (Kadcyla), inotuzumab ozogamicin (Besponsa), brentuximab vedotin (Adcetris) and others	[28]
Immunotoxins	Catalytic subunits of natural toxins (pseudomonas exotoxin, diphtheria toxin)		Cytosol	Three immunotoxins have been approved for clinical use, with one of them—IL-3 fused to a fragment of diphtheria toxin (Tagraxofusp)—currently used in clinical settings	[27]
p-PROTACs (peptide-based proteolysis-targeting chimaeras)		Intracellular proteins of interest (e.g., protein kinases, estrogen receptor, Tau-protein)	Proteasome		[64]
LYTACs (lysosome-targeting chimaeras)		Extracellular and membrane-associated proteins of interest (e.g., EGFR, PD-L1, transferrin receptor-1)	Lysosome		[58]
AbTACs (antibody-based targeting chimaeras)		Cell surface protein of interest (e.g., PD-L1)	Lysosome		[65]
NLS-modified monoclonal antibody-DTPA	Auger electron emitter (^111^In)		Nucleus		[66]
Nucleolin-binding F3-peptide-DTPA	Auger electron emitter (^111^In)		Nucleus		[67]
Acid-activated cell-penetrating peptide (CPP) with NLS	Chemotherapeutics (camptothecin)		Nucleus		[48]
MNTs (modular nanotransporters)	Auger electron emitters (^111^In, ^125^I, ^67^Ga), α-emitter ^211^At, photosensitizers		Nucleus		[68]
Cell-targeting peptide functionalized CPP with NLS and oligohistidine	Plasmid DNA (pDNA)		Nucleus		[69]
CPP with ER retention sequence	Interleukin-24		ER		[70]
Self-assembling peptides	Histone protein H2B, chemotherapeutics (doxorubicin)		Mitochondria		[71,72]
Histone-mediated transduction	pDNA		Nucleus		[73]
Peptide/pDNA self-assembling complexes	pDNA		Mitochondria		[69]

EGFR—epidermal growth factor receptor, PD-L1—programmed death ligand-1, NLS—nuclear localization signal, ER—endoplasmic reticulum-

## Data Availability

Not applicable.

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
