# Peer review of "Prospects of Using Protein Engineering for Selective Drug Delivery into a Specific Compartment of Target Cells"

_pharmaceutics, 2023, doi:10.3390/pharmaceutics15030987_

Round 1

Reviewer 1 Report

The review article titled “Prospects of using protein engineering for selective drug delivery into a specific compartment of target cells“ discusses the successful use of various proteins, including natural polypeptide hormones, synthetic analogues, antibodies, enzymes, and other drugs for treating diseases. They highlights that most therapeutic targets are located inside the cell and are regulatory macromolecules, making it difficult to develop small molecules that can specifically affect protein interactions without causing side effects, while multifunctional proteins that can interact with almost any target, but these proteins cannot easily penetrate the desired compartment of the cell. The review covers the scope of application of these artificial constructs, obstacles faced during transport to the intracellular compartment of target cells, and ways to overcome these challenges. The authors also summarized protein based systems for subcellular targeted delivery table with demonstrated category of delivery system, payloads, and subcellular destination. In general, The paper is well written, with clear objectives. However, few comments should be addressed before publication.

Although Figure 1 describes pathways for approaching intracellular drug targets, the article lacks an illustrative description of protein engineering strategies to overcome the extra-/intra-cellular barriers for selective drug delivery into a specific compartment of target cells.

The following papers are highly associated with the present work, which would fit well in the section discussing extra-/intra-cellular barriers for subcellular targeted delivery.

1. https://doi.org/10.3390/biomimetics7030126

Author Response

Point 1: Although Figure 1 describes pathways for approaching intracellular drug targets, the article lacks an illustrative description of protein engineering strategies to overcome the extra-/intra-cellular barriers for selective drug delivery into a specific compartment of target cells.

Response 1: 

Thank you for the review of our manuscript and for this reasonable suggestion. We added Figure 4, which describes protein engineering strategies to overcome the extra-/intra-cellular barriers for selective drug delivery into a specific compartment of target cells, into the “5.3. Combining transport functions in a single polypeptide chain” section.

Point 2: The following papers are highly associated with the present work, which would fit well in the section discussing extra-/intra-cellular barriers for subcellular targeted delivery.

https://doi.org/10.3390/biomimetics7030126

Response 2:

Thank you for the suggestion. We added the reference.

Reviewer 2 Report

In the reviewProspects of using protein engineering for selective drug delivery into a specific compartment of target cells’ by Andrey A. Rosenkranz and Tatiana A. Slastnikova, the authors discuss therapeutic proteins capable of interacting with almost any target obtained by modern technologies. This review considers the scope of application of such artificial constructs; the obstacles met on the way of their transport to the specified intracellular compartment of target cells after their systemic blood-stream administration and means to overcome these difficulties.

The manuscript needs major revision as follows

Comment 1. Introduction section needs development as the fourth paragraph (lines 68-81) addresses drugs attributed to the fourth group, which penetrate into cells and perform their effects on intracellular processes in specified cell compartments. However, it is not clear from the text in this paragraph whether peptides/polypeptides (proteins) or their artificial analogues used to functionalize new kinds of nanoparticles as a mean of drug delivery are the aim of this review or protein drugs that can penetrate and transported into cell compartments by recombination and artificial constructs as stated in the title and abstract as focus of this review or both of them. The aim should be clear.

Comment 2. Introduction have no information regarding modern technologies used to design penetrative recombinant therapeutic proteins and peptides into specified cell compartments and examples of these protein drugs.

As an example: Intracellular delivery of protein drugs with an autonomously lysing bacterial system that succeed in targeted killing of cancer cells, Cell-Penetrating Peptides, etc.

[1] Raman V, Van Dessel N, Hall CL, Wetherby VE, Whitney SA, Kolewe EL, Bloom SMK, Sharma A, Hardy JA, Bollen M, Van Eynde A, Forbes NS. Intracellular delivery of protein drugs with an autonomously lysing bacterial system reduces tumor growth and metastases. Nat Commun. 2021 Oct 21;12(1):6116. doi: 10.1038/s41467-021-26367-9.

[2] Xie J, Bi Y, Zhang H, Dong S, Teng L, Lee RJ and Yang Z (2020) Cell-Penetrating Peptides in Diagnosis and Treatment of Human Diseases: From Preclinical Research to Clinical Application. Front. Pharmacol. 11:697. doi: 10.3389/fphar.2020.00697

[3] Jiaqi Fu, Changmin Yu, Lin Li and Shao Q. Yao Intracellular Delivery of Functional Proteins and Native Drugs by Cell-Penetrating Poly(disulfide)s J. Am. Chem. Soc. 2015, 137, 37, 12153–12160

[4] Sang-Eun Bae, Soo Kyung Lyu, Ki-Jung Kim, Hee Joo Shin, Hyockman Kwon and Seong Huh. Intracellular delivery of a native functional protein using cell-penetrating peptide functionalized cubic MSNs with ultra-large mesopores.  

Issue 21, 2018 Journal of Materials Chemistry B

[5] Cytosolic targeting of drug therapeutics; delivery of Human growth hormone in Thermosensitive micelles, Micellar cyclotriphosphazenes in rabbits.   Toti US, Moon SH, Kim HY, Jun YJ, Kim BM, Park YM, Jeong B, Sohn YS. Thermosensitive and biocompatible cyclotriphosphazene micelles. J Control Release. 2007 May 14;119(1):34-40. doi: 10.1016/j.jconrel.2007.01.003.

Comment 3. The last paragraph in introduction (lines 84-85), stated that ‘Therefore, an attractive feature of the approach is the ability to include an active principle directly into the developed construct’ this comes rather suddenly, as the authors did not mention the approaches of developing artificial constructs in the first place to state inclusion of active amino acids domains.

Comment 4. The introduction lacks clear detailing of the aim of the review as introductory text to the next points.

Comment 5. Section 2.1. Cell specificity

The authors mentioned natural hormones and cytokines, blocking antibodies as examples of drugs that bind to cell surface receptors. Are there examples for protein drugs that target proteomes of individual cell types.

Comment 6. Section 2.2. Subcellular specificity

The authors mentioned many examples of drugs that are not proteins or peptides, which is not the aim of the review described in abstract. 

Comment 7. Table 2 needs reformatting

Again the confusion, is the target of the review therapeutic proteins engineered to penetrate and target cellular compartments OR using proteins as delivery system to other chemical drugs or both of them.

Comment 8. Lines 341-350: have no references

Comment 9. Lines 469-476: have no references

Comment 10. 8.1. Endosomes acidification and protein drug delivery

This idea should be made clearer

Endosomal escape remains a major issue for developing an efficient and targeted delivery system for therapeutic applications. The endosomal-lysosomal system is made up of a set of intracellular membranous compartments that dynamically interconvert, which is comprised of early endosomes, recycling endosomes, late endosomes, and the lysosome.  Lysosomes are the terminal compartment of the endocytic pathway. Following the internalization process, drugs must escape into the cell cytoplasm for evading degradation by hydrolytic enzymes present in the lysosomes. 

Comment 11. Lines 636- 646 lack references  

Comment 12. Lines 647-648 and lines 666-: For many years, a promising research direction that uses cell-penetrating peptides (CPP) to deliver drugs into the cytosol has been developing. Another option for delivery to the cytosol is to use fragments of a number of bacterial toxins that are specialized for this process. These approaches should be mentioned with other approaches in introduction section.

Comment 13. Lines 698-700: Various NLSs are often included in constructs based on nanoparticles, polymers, micelles, and various polypeptides, including natural proteins, antibodies and their derivatives, and synthetic peptides for targeted delivery of the active agent into the nucleus. This approach should be mentioned with other approaches in introduction section.

Comment 14. Lines 801- 808: these lines concerning artificial protein molecules should be detailed in introduction and aim of the review.  

Minor comments

Please revise abbreviations throughout the manuscript and ensure the presence of their full names when first mentioned only.

Ensure when stating results of a study that the reference is cited in the paragraph of results.

Author Response

Point 1: Comment 1. Introduction section needs development as the fourth paragraph (lines 68-81) addresses drugs attributed to the fourth group, which penetrate into cells and perform their effects on intracellular processes in specified cell compartments. However, it is not clear from the text in this paragraph whether peptides/polypeptides (proteins) or their artificial analogues used to functionalize new kinds of nanoparticles as a mean of drug delivery are the aim of this review or protein drugs that can penetrate and transported into cell compartments by recombination and artificial constructs as stated in the title and abstract as focus of this review or both of them. The aim should be clear.

Response 1: Thank you for this reasonable suggestion. We added necessary clarification to both the Abstract and the Introduction section, and directly stated the aim of the Review at the end of the Introduction section as follows: “The current field of multifunctional protein-engineered therapeutics intended for selective drug delivery, their prospects, and their limitations are summarized in this review. These constructs can carry either low molecular weight drugs or therapeutic amino acid sequences.”

Point 2: Comment 2. Introduction have no information regarding modern technologies used to design penetrative recombinant therapeutic proteins and peptides into specified cell compartments and examples of these protein drugs.

As an example: Intracellular delivery of protein drugs with an autonomously lysing bacterial system that succeed in targeted killing of cancer cells, Cell-Penetrating Peptides, etc.

[1] Raman V, Van Dessel N, Hall CL, Wetherby VE, Whitney SA, Kolewe EL, Bloom SMK, Sharma A, Hardy JA, Bollen M, Van Eynde A, Forbes NS. Intracellular delivery of protein drugs with an autonomously lysing bacterial system reduces tumor growth and metastases. Nat Commun. 2021 Oct 21;12(1):6116. doi: 10.1038/s41467-021-26367-9.

[2] Xie J, Bi Y, Zhang H, Dong S, Teng L, Lee RJ and Yang Z (2020) Cell-Penetrating Peptides in Diagnosis and Treatment of Human Diseases: From Preclinical Research to Clinical Application. Front. Pharmacol. 11:697. doi: 10.3389/fphar.2020.00697

[3] Jiaqi Fu, Changmin Yu, Lin Li and Shao Q. Yao Intracellular Delivery of Functional Proteins and Native Drugs by Cell-Penetrating Poly(disulfide)s J. Am. Chem. Soc. 2015, 137, 37, 12153–12160

[4] Sang-Eun Bae, Soo Kyung Lyu, Ki-Jung Kim, Hee Joo Shin, Hyockman Kwon and Seong Huh. Intracellular delivery of a native functional protein using cell-penetrating peptide functionalized cubic MSNs with ultra-large mesopores.  

Issue 21, 2018 Journal of Materials Chemistry B

[5] Cytosolic targeting of drug therapeutics; delivery of Human growth hormone in Thermosensitive micelles, Micellar cyclotriphosphazenes in rabbits.   Toti US, Moon SH, Kim HY, Jun YJ, Kim BM, Park YM, Jeong B, Sohn YS. Thermosensitive and biocompatible cyclotriphosphazene micelles. J Control Release. 2007 May 14;119(1):34-40. doi: 10.1016/j.jconrel.2007.01.003.

Response 2: Thank you for this suggestion. We added the mentioned references [1] – [4] and a few more:

“Many drug delivery vehicles that facilitate penetration into cells, regardless of cell type, are in fact close to this group. Such delivery systems include, for example, cell-penetrating peptides (CPP) [2], bacteria-based delivery systems [3], cell-penetrating poly(disulfide)s [4], mesoporous silica nanoparticles [5, 6], micelles [7] and many others.”

Unfortunately, the article # [5] (Toti US, Moon SH, Kim HY, Jun YJ, Kim BM, Park YM, Jeong B, Sohn YS, 2007, Thermosensitive and biocompatible cyclotriphosphazene micelles. Journal of Controlled Release, 119(1), 34-40, doi: 10.1016/j.jconrel.2007.01.003) you suggested is not related to intracellular delivery.

Point 3: Comment 3. The last paragraph in introduction (lines 84-85), stated that ‘Therefore, an attractive feature of the approach is the ability to include an active principle directly into the developed construct’ this comes rather suddenly, as the authors did not mention the approaches of developing artificial constructs in the first place to state inclusion of active amino acids domains.

Response 3: Thank you for careful reading and pointing at this issue. To improve the clarity of the logic of these paragraphs, we have rearranged them slightly and specified the approach we are considering:

“Therefore, an attractive feature of the protein-based approach is its ability not only to deliver convenient low-molecular drugs into the most appropriate cell compartment, but also to incorporate an active principle directly into the developed construct. In the future, this may lead to the precise regulation of a pathologically altered process adapted for a particular patient. There are many obstacles to the implementation of such a direction, concerning both the existing barriers to the delivery of a suitable drug to the right place of the right cell, and the lack of our knowledge. Nevertheless, it is already possible to try to evaluate the necessary properties of such structures and the conditions that would allow them to be obtained. This direction is all the more important, since 75-80% of protein targets are intracellular proteins that lack obvious sites for the action of small-sized drugs [23]. Alternatively, gene therapy approaches [24] as well as mRNA delivery [25] are also considered as means to provide the appearance of the right protein in the right cell. In this case, it is necessary to ensure effective delivery of DNA or RNA to the nucleus or cytosol, respectively, of the target cell. The delivery of nucleic acids for therapeutic purposes is a separate area that is generally outside the scope of this review.

The current field of multifunctional protein-engineered therapeutics intended for selective drug delivery into a specific compartment of target cells, their prospects, and their limitations are summarized in this review. These constructs can carry either low molecular weight drugs or therapeutic amino acid sequences.”

Point 4: Comment 4. The introduction lacks clear detailing of the aim of the review as introductory text to the next points.

Response 4: Thank you for the valuable suggestion. We added the clearly detailed aim of the Review at the end of the Introduction section as follows: “The current field of multifunctional protein-engineered therapeutics intended for selective drug delivery into a specific compartment of target cells, their prospects, and their limitations are summarized in this review. These constructs can carry either low molecular weight drugs or therapeutic amino acid sequences.”

Point 5: Comment 5. Section 2.1. Cell specificity

The authors mentioned natural hormones and cytokines, blocking antibodies as examples of drugs that bind to cell surface receptors. Are there examples for protein drugs that target proteomes of individual cell types.

Response 5: Thank you for your question, this approach is under extensive development now. As an example of protein drug targeted at proteomes of individual cell types can serve modular nanotransporter for targeted delivery of p21 protein fragment (already included into nanotransporter molecule) into the nuclei of EGFR-expressing cancer cells [Kamaletdinova, T. R.; Rosenkranz, A. A.; Ulasov, A. V.; Khramtsov, Y. V.; Tsvetkova, A. D.; Georgiev, G. P.; Sobolev, A. S. Modular Nanotransporter with P21 Fragment Inhibits DNA Repair after Bleomycin Treatment. Dokl. Biochem. Biophys. 2018, 479 (1), 95-97].

Point 6: Comment 6. Section 2.2. Subcellular specificity

The authors mentioned many examples of drugs that are not proteins or peptides, which is not the aim of the review described in abstract. 

Response 6: Thank you for careful reading. We mentioned examples of cargo drugs that are not proteins or peptides, as the protein-based delivery systems for these drugs are also the aim of this Review. To make it clear, we changed the abstract a little, to state the scope of the review (“This review considers the scope of application of such artificial constructs for targeted delivery of both protein-based and traditional low molecular weight drugs”) clearer. Also, we directly stated the aim of the Review at the end of the Introduction section as follows: “The current field of multifunctional protein-engineered therapeutics intended for selective drug delivery into a specific compartment of target cells, their prospects, and their limitations are summarized in this review. These constructs can carry either low molecular weight drugs or therapeutic amino acid sequences.”

Point 7: Comment 7. Table 2 needs reformatting

Again the confusion, is the target of the review therapeutic proteins engineered to penetrate and target cellular compartments OR using proteins as delivery system to other chemical drugs or both of them.

Response 7: Thank you for careful reading. We changed the abstract a little, to state the scope of the review clearer (the added text is underlined):“This review considers the scope of application of such artificial constructs for targeted delivery of both protein-based and traditional low molecular weight drugs”. Also, we directly stated the aim of the Review at the end of the Introduction section as follows: “The current field of multifunctional protein-engineered therapeutics intended for selective drug delivery into a specific compartment of target cells, their prospects, and their limitations are summarized in this review. These constructs can carry either low molecular weight drugs or therapeutic amino acid sequences.”

Point 8: Comment 8. Lines 341-350: have no references

Response 8: Thank you for your comment. We added appropriate references:

“The principal possibilities of obtaining such information have recently been demonstrated by the example of changes in protein phosphorylation and changes in their intracellular localization after the addition of epidermal growth factor [34]. At the same time, more than 7,000 proteins and 11,000 phosphorylation sites were monitored simultaneously in experiments on HeLa cell culture. Comparison with a similar experiment conducted in vivo showed both the presence of a correlation with data on cell culture and some differences that are anticipated when studying regulation systems at different levels of the living [34]. Such approaches, in addition to the already available detailed work on the study of many regulatory pathways, make it possible to assess how the cell as a whole reacts to the received stimulus.”

Point 9: Comment 9. Lines 469-476: have no references

Response 9: Thank you for your comment. We added appropriate references:

“Recognition of the cell type, transport of macromolecules into cells, and signal transduction via intracellular pathways all are largely carried out through the interaction or modification of proteins. The variety of amino acid sequences makes it possible to combine the necessary transport functions with therapeutic ones within a single molecule encoded by a single gene [68, 144, 145]. This makes it possible to easily obtain a reproducible product and ensures the scalability of its production. For this kind of drug systems, it is necessary to preserve the properties that ensure both transport to the desired part of the target cell and the performance of the delivered drug.”

Point 10: Comment 10. 8.1. Endosomes acidification and protein drug delivery

This idea should be made clearer

Endosomal escape remains a major issue for developing an efficient and targeted delivery system for therapeutic applications. The endosomal-lysosomal system is made up of a set of intracellular membranous compartments that dynamically interconvert, which is comprised of early endosomes, recycling endosomes, late endosomes, and the lysosome.  Lysosomes are the terminal compartment of the endocytic pathway. Following the internalization process, drugs must escape into the cell cytoplasm for evading degradation by hydrolytic enzymes present in the lysosomes. 

Response 10: Thank you very much for your comment. We added into section 8.1 the following text: “This pathway is a highly dynamic endocytic membrane system, which is comprised of early endosomes, recycling endosomes, late endosomes/multivesicular bodies, and lysosomes.”

Point 11: Comment 11. Lines 636- 646 lack references  

Response 11: Thank you for your comment. We added appropriate references:

“There are rather many amino acid sequences that can provide a disturbance of the lipid bilayer integrity in a slightly acidic environment. Many of them are used to study the possibility of effective delivery of drugs to the cytosol [195]. For drug delivery into cytosol of a target cell by receptor-mediated endocytosis, it is critical that the interaction of such a sequence with the lipid bilayer and pore formation or membrane rupture occur in a slightly acidic environment at pH 6.5-5.5. The peptide sequence interacting with the phospholipid bilayer under neutral conditions will undergo nonspecific embedding into the plasma membrane of any cell. It is clear that the specificity of drug delivery to a specified cell type is hardly achievably in this case. At the same time, if such a sequence disrupts the lipid bilayer at pH 5 and below, the delivered drug will be hydrolyzed in lysosomes [196].”

Point 12: Comment 12. Lines 647-648 and lines 666-: For many years, a promising research direction that uses cell-penetrating peptides (CPP) to deliver drugs into the cytosol has been developing. Another option for delivery to the cytosol is to use fragments of a number of bacterial toxins that are specialized for this process. These approaches should be mentioned with other approaches in introduction section.

Response 12: Thank you for your comment. We added in the Introduction section: “Many drug delivery vehicles that facilitate penetration into cells, regardless of cell type, are in fact close to this group. Such delivery systems include, for example, cell-penetrating peptides (CPP) [2],…”

and

“Many of these therapeutics are based on the synthesis of nanoparticles decorated with various functional components, including various ligands for internalized surface receptors, e.g., proteins, carbohydrates and low molecular weight substances, CPPs, fragments of bacterial toxins for delivery to the cytosol, nuclear localization signals (NLS) for delivery to the nucleus and polypeptide sequences for transport into cell organelles.”

Point 13: Comment 13. Lines 698-700: Various NLSs are often included in constructs based on nanoparticles, polymers, micelles, and various polypeptides, including natural proteins, antibodies and their derivatives, and synthetic peptides for targeted delivery of the active agent into the nucleus. This approach should be mentioned with other approaches in introduction section.

Response 13: Thank you for your comment. We added in the Introduction section:

“Many of these therapeutics are based on the synthesis of nanoparticles decorated with various functional components, including various ligands for internalized surface receptors, e.g., proteins, carbohydrates and low molecular weight substances, CPPs, fragments of bacterial toxins for delivery to the cytosol, nuclear localization signals (NLS) for delivery to the nucleus and polypeptide sequences for transport into cell organelles.”

Point 14: Comment 14. Lines 801- 808: these lines concerning artificial protein molecules should be detailed in introduction and aim of the review.  

Response 14: Thank you for your comment. We added to Introduction section:

“The current field of multifunctional protein-engineered therapeutics intended for selective drug delivery into a specific compartment of target cells, their prospects, and their limitations are summarized in this review. These constructs can carry either low molecular weight drugs or therapeutic amino acid sequences.”

Point 15: Minor comments

Please revise abbreviations throughout the manuscript and ensure the presence of their full names when first mentioned only.

Response 15: We are grateful for this remark and have removed unnecessary abbreviations with the exception of tables and figure legends.

Reviewer 3 Report

Overall, this review article was informative, broad in its scope, and provided unique insights into drug delivery and drug selectivity for target cells. However, I found the organization of the manuscript, as well as the language to need extensive editing and improvement. In particular, I couldn't understand the logic for why the topics/sections were introduced and arranged as they appeared in the manuscript. This also contributed to repetition in content in places, as well as a manuscript that was long and that could stand to be significantly shortened. 

Specific Comments:

Lns 15-16: I suppose the statement that the vast majority of therapeutic targets are regulatory macromolecules is correct, but it seemed oddly phrased.

Lns 38-39: again, I didn't understand what the authors meant by "relative limitation of chemical space they occupy, which determines the exhaustibility of the approach."  

Figure 1: I found this to be overly simplistic and lack the nuances discussed in the text. For example, non-target cells may also contain the membrane proteins and receptors of the target cells. I would suggest to remove the non-target cells from this figure to avoid this confusion.

Lns 101-110: some of this section seems repetitive with the introduction.

Lns 126-129: the authors mention that GPCRs may provide an avenue for specific drug delivery to cells. I'd imagine that the rate of uptake via this route might be inefficient due to the small number of receptors generally expressed on the cell surface and that their internalization may be relatively slow. Can the authors comment on this and provide some justification for the use of GPCRs as a viable drug delivery system?

Ln 158: what do the authors mean by "individual differences"? Perhaps the authors could be a little more precise in the information they wish to convey with this phrase. I'd also suggest moving up the discussion of the effect of PTMs and protein interactions (introduced in Lns 337-340) to this section, since these can have a determining effect on structure, function, and localization, all of which can effect drug action.

Table 2: I was curious how many of the delivery systems described in this table and the paragraphs related to this table are currently being used in the clinic. Perhaps the authors could give an indication of this in the table?

Lns 253-254: could the authors clarify what they mean by a greater increase in size? Do they mean above 100nm?

Lns 292-293: I'd suggest to define EPR for the reader.

Lns 305-312: the authors might want to discuss the therapeutic potential of transcytosis for the delivery of agents across the BBB. 

Author Response

Point 1: Overall, this review article was informative, broad in its scope, and provided unique insights into drug delivery and drug selectivity for target cells. However, I found the organization of the manuscript, as well as the language to need extensive editing and improvement. In particular, I couldn't understand the logic for why the topics/sections were introduced and arranged as they appeared in the manuscript. This also contributed to repetition in content in places, as well as a manuscript that was long and that could stand to be significantly shortened. 

Response 1: Thank you for your comment. We added following text in the Introduction section to make the logic of manuscript architecture easier to understand: “Cell and intracellular specificity, as the keys to selective targeting, will be discussed first. Then we shed light on extracellular and intracellular barriers for therapeutics to overcome on their way to the intracellular target. This is followed by a review of internalizable protein-based delivery systems and their intracellular fate after cell entry.” Also we added following text in the Section 4: “In-depth description of the pathways for targeting of each individual intracellular organelle is done in the Section 8. Branching delivery pathways for endocytosed drugs.” In order to improve the organization of the manuscript we re-arranged some parts of it: 1) we moved up the discussion of the effect of PTMs and protein interactions: “Moreover, approaches to the study of the spatio-temporal regulation of the proteome in the cell and its posttranslational regulatory modifications are beginning to appear [34]. The principal possibilities of obtaining such information have recently been demonstrated by the example of changes in protein phosphorylation and changes in their intracellular localization after the addition of epidermal growth factor [34]. At the same time, more than 7,000 proteins and 11,000 phosphorylation sites were monitored simultaneously in experiments on HeLa cell culture. A comparison with a similar experiment conducted in vivo showed both the presence of a correlation with data on cell culture and some differences that are anticipated when studying regulation systems at different levels of the living [34]. Such approaches, in addition to the already available detailed work on the study of many regulatory pathways, make it possible to assess how the cell as a whole reacts to the received stimulus.” to the section 2.1. Cell specificity; 2) we deleted part of section 4. (Intracellular targets and barriers on the way to them): “Transport of macromolecules from the cytosol into the nucleus is controlled by the nuclear pore complex (NPC) that crosses outer and inner nuclear membranes. The NPC is a selectively permeable barrier that allows both passive diffusion of small molecules and active transport of macromolecules more than 40 kDa [124].”; 3) we moved down the following text: “There is also a channel between the ER and the cytosol through which cellular proteins can be transported from one compartment to another. For transport from the cytosol to the ER, it works mainly for newly synthesized proteins that are designed to function in the ER or outside the cell. This transport is mediated by the Sec61 translocon, which interacts with the ribosomes of the rough ER. Thousands of secretory and membrane proteins follow this pathway [271, 272]. The Sec61 translocon likely also works in the opposite direction for ER-associated protein degradation (ERAD) by transporting misfolded proteins into the cytosol [273, 274]. Apparently, there are several retrotranslocons that provide reverse transport of misfolded and unfolded proteins, from the ER to the cytosol, including Sec61 [274], the Hrd1-Hrd3 complex [275], Derlin family members (Der1-3) [276], and the Hrd1-Der1 complex [277]. The process is closely related to the ubiquitination and degradation of such proteins in proteasomes [278, 279]. In addition, Sec61 seems to mediate the transport of a number of receptors after internalization of ligand-receptor complexes and vesicular transport of some of them from the endosomes to the TGN and further into the cytosol and cell nucleus. Such a process has been noted for one of the most frequently overexpressed receptors in cancer, EGFR [280], and for some other tyrosine kinases [281]. The throughput of this additional retrograde transport pathway of endocytosed macromolecules is much inferior to transport into lysosomes or recycling to the cell surface. The presence of this macromolecule transport pathway is necessary for some bacterial toxins, which, after leaving the ER, are folded in the cytosol and not cleaved by the proteasome [258, 282-284].” to the section 8.5. Vesicular transport to intracellular targets.

We have also edited the language of the manuscript.

Specific Comments:

Point 2: Lns 15-16: I suppose the statement that the vast majority of therapeutic targets are regulatory macromolecules is correct, but it seemed oddly phrased.

Response 2: Thank you for careful reading. We re-phrased this statement “Meanwhile the vast majority of therapeutic targets do locate inside the cell and are regulatory macromolecules.” as follows: “Meanwhile the vast majority of therapeutic targets, which are usually regulatory macromolecules, locate inside the cell.”

Point 3: Lns 38-39: again, I didn't understand what the authors meant by "relative limitation of chemical space they occupy, which determines the exhaustibility of the approach."

Response 3: Thank you for careful reading. We deleted this sentence.

Point 4: Figure 1: I found this to be overly simplistic and lack the nuances discussed in the text. For example, non-target cells may also contain the membrane proteins and receptors of the target cells. I would suggest to remove the non-target cells from this figure to avoid this confusion.

Response 4: Thank you for this comment. We agree that Figure 1 simplifies the reality and lacks some nuances discussed in text (as, however, any schematic representation does). Meanwhile, we are sure that the presence of non-target cells is essential for understanding, and their exclusion will result in further simplification. Instead, we have added some receptors to non-target cells.

Point 5: Lns 101-110: some of this section seems repetitive with the introduction.

Response 5: Thank you for thorough reading. We changed the aforementioned lines as follows: “This kind of drugs is presented by their natural agonists as well as natural and synthetic antagonists, including blocking antibodies” to avoid any repetitions.

Point 6: Lns 126-129: the authors mention that GPCRs may provide an avenue for specific drug delivery to cells. I'd imagine that the rate of uptake via this route might be inefficient due to the small number of receptors generally expressed on the cell surface and that their internalization may be relatively slow. Can the authors comment on this and provide some justification for the use of GPCRs as a viable drug delivery system?

Response 6: Thank you for your reasonable question. Indeed, the expression of most GCPR is low. However, our own experience with GPCR MCR1 had shown that when 5-10 thousands receptors are expressed per cell, the efficiency of photosensitizers delivered to the cell nucleus of melanoma cells via this receptor increases by two orders of magnitude in the absence of any effect on cells with negligible expression of MC1R (DOI: 10.1096/fj.02-0888fje). The expression of many GPCRs is often increased in malignancies (Dorsam, R. T., & Gutkind, J. S. (2007). G-protein-coupled receptors and cancer. Nature reviews cancer, 7(2), 79-94, DOI: 10.1038/nrc2069; Lappano, R., & Maggiolini, M. (2011). G protein-coupled receptors: novel targets for drug discovery in cancer. Nature reviews Drug discovery, 10(1), 47-60, DOI: 10.1038/nrc206; Chaudhary, P. K., & Kim, S. (2021). An insight into GPCR and G-proteins as cancer drivers. Cells, 10(12), 3288, DOI: 10.3390/cells10123288). Therefore, many GPCRs are often considered as promising targets for anti-cancer therapy. In addition, GPCRs are only a part of surface receptors, the expression of many of which is significantly higher than for most GCPR. We do not consider GCPR alone as an avenue for intracellular delivery.

 Point 7: Ln 158: what do the authors mean by "individual differences"? Perhaps the authors could be a little more precise in the information they wish to convey with this phrase. I'd also suggest moving up the discussion of the effect of PTMs and protein interactions (introduced in Lns 337-340) to this section, since these can have a determining effect on structure, function, and localization, all of which can effect drug action.

Response 7: Thank you for thorough reading. We clarified what we mean by individual differences changing this to “personal individual differences”. We also moved up the discussion of the effect of PTMs and protein interactions: “Moreover, approaches to the study of the spatio-temporal regulation of the proteome in the cell and its posttranslational regulatory modifications are beginning to appear [34]. The principal possibilities of obtaining such information have recently been demonstrated by the example of changes in protein phosphorylation and changes in their intracellular localization after the addition of epidermal growth factor [34]. At the same time, more than 7,000 proteins and 11,000 phosphorylation sites were monitored simultaneously in experiments on HeLa cell culture. Comparison with a similar experiment conducted in vivo showed both the presence of a correlation with data on cell culture and some differences that are anticipated when studying regulation systems at different levels of the living [34]. Such approaches, in addition to the already available detailed work on the study of many regulatory pathways, make it possible to assess how the cell as a whole reacts to the received stimulus.” to this section.

Point 8: Table 2: I was curious how many of the delivery systems described in this table and the paragraphs related to this table are currently being used in the clinic. Perhaps the authors could give an indication of this in the table?

Response 8: Thank you for this valuable suggestion. We added the column to the Table 2, where we indicated which of the described delivery systems are currently being used in clinic.

Point 9: Lns 253-254: could the authors clarify what they mean by a greater increase in size? Do they mean above 100nm?

Response 9: Thank you for your reasonable question. We added the following clarification to the text of the manuscript:

“The efficiency of this uptake depends not only on the size, but also on the type of nano-particles, their surface charge, shape, and properties of their surface. The nanoparticle uptake by macrophages shows an exponential dependence on their size in the range from 30 to 300 nm [79].”

Point 10: Lns 292-293: I'd suggest to define EPR for the reader.

Response 10: Thank you for the suggestion, we modified the sentences as follows (the added text is underlined): “Many tumors are characterized by defective architecture of the vessels and lymphatic system, characterized by leaky vasculature and defective lymphatic drainage. For a number of therapeutic macromolecules and nanoparticles, this leads to enhanced permeability and retention (EPR) [87, 88], resulting in their tumoritropic accumulation.” in order to define EPR for the reader more clearly.

Point 11: Lns 305-312: the authors might want to discuss the therapeutic potential of transcytosis for the delivery of agents across the BBB. 

Response 11: Thank you for the suggestion. We added: “In the central nervous system, where endothelial cells are characterized by unusually highly restricted molecular exchange and very tight junctions forming the blood-brain barrier, delivery of therapeutics becomes even more challenging. Receptor mediated transcytosis relying on binding to specific receptors, e.g., transferrin receptors or insulin receptors, is supposed to be a prospective approach to overcome this issue [99].”

Round 2

Reviewer 2 Report

The authors did a great job at improving their manuscript. I suggest it can be now accepted for publication

Reviewer 3 Report

The authors have responded to all of my previous comments and I consider the manuscript significantly improved in terms of its organization and clarity. I have no further suggestions.